# Novel Identification of the Collection of Pathogenic Fungal Species *Verticillium* with the Development of Species-Specific SSR Markers

**DOI:** 10.3390/pathogens12040535

**Published:** 2023-03-29

**Authors:** Taja Jeseničnik, Anela Kaurin, Zarja Grgič, Sebastjan Radišek, Jernej Jakše, Nataša Štajner

**Affiliations:** 1Department of Agronomy, Biotechnical Faculty, University of Ljubljana, 1000 Ljubljana, Slovenia; 2Plant Protection Department, Slovenian Institute of Hop Research and Brewing, 3310 Žalec, Slovenia

**Keywords:** classification, LAMP, multiplex PCR, simplex PCR, SSR marker, *Verticillium*

## Abstract

The genus *Verticillium* is a group of ascomycete fungi that includes several pathogenic plant species. In 2011, a new taxonomic classification, proposed by Inderbitzin and coworkers (2011), re-defined the genus as *Verticillium* sensu stricto. The objective of our study was the re-classification of the fungal species held in the culture collection in the Slovenian Institute of Hop Research and Brewing in accordance with the newly established taxonomy. With the PCR marker system proposed by Inderbitzin and coworkers in 2011, we re-classified 88 *Verticillium* isolates out of the 105 samples that are held in the institute’s bank, which were obtained from different geographic locations in Europe, North America, and Japan, and from different host plants, including alfalfa, cotton, hop, olive, potato, and tomato. However, the PCR marker for the *V. dahliae* identification proved to be less specific, and it resulted in the positive amplification of *Gibellulopsis nigrescens*, *V. isaacii,* and *V. longisporum*. To enable the accurate distinction of the fungi, the SSR and LAMP markers were added to the analyses. The 12 newly identified SSR markers, which were used in simplex PCR reactions or in combination, enabled the accurate identification of all included *Verticillium* isolates and could potentially be used as biomarkers for rapid and easy species identification.

## 1. Introduction

The genus *Verticillium* is a group of ascomycete filamentous fungi that includes plant-pathogenic species that infect the vasculature of many agricultural crops, leading to large economic losses worldwide [1,2,3,4,5,6,7]. In 2011, a new taxonomic classification of the genus was established, introducing the reduced genus *Verticillium* sensu stricto, which comprises 10 plant-pathogenic fungal species: *Verticillium albo-atrum*, *Verticillium alfalfae*, *Verticillium dahliae*, *Verticillium isaacii*, *Verticillium klebahnii*, *Verticillium longisporum, Verticillium nonalfalfae*, *Verticillium nubilum*, *Verticillium tricorpus,* and *Verticillium zaregamsianum* [2]. On the basis of extensive molecular analyses, *Verticillium nigrescens* and *Verticillium theobromae* were moved to other genera [7].

*Verticillium* sensu stricto species are soil-borne fungi with hyaline mycelia. Hyphae and conidia are mostly haploid, and conidia are produced on long philaides [4,8,9]. One of the features of *Verticillium* sensu stricto species is the formation of resting structures, by which the species remain dormant in the soil for long periods, even without a host plant [2,4,9,10]. The three types of resting structures include resting mycelia, microscleorotia, and clamidospores, with the species *V. albo-atrum*, *V. isaacii*, *V. klebahnii,* and *V. tricorpus* forming more than one type of resting structure [2]. The *Verticillium* sensu stricto species also differ in pathogenicity: the less virulent and highly virulent fungal isolates of *V. nonalfalfae* cause mild and lethal wilting symptoms in hop plants, respectively; and infection with defoliating and non-defoliating pathotypes of *V. dahliae* manifests as wilting with or without defoliation on cotton and olive plants [11]. The species also differ in host range, with *V. dahliae* capable of infecting plants from 14 plant families, whereas *V. alfalfae* only infects lucerne [2,5,12,13,14,15,16].

In a number of recent studies, it has become increasingly apparent that identification based on morphological and physiological characteristics is not always unambiguous, since the characteristics used for identification, including resting structures and yellow hyphal pigmentation, are unstable and may disappear in laboratory cultures. Thus, some species lack distinguishing morphological characteristics [17]. For example, the *V. dahliae* species is morphologically difficult to distinguish from the *V. longisporum* lineages; *V. alfalfae* and *V. nonalfalfae* are morphologically very similar to the distantly related *V. albo-atrum* species; and *V. isaacii* and *V. klebahnii* are morphologically indistinguishable from *V. tricorpus* [2]. Thus, the accurate and reliable identification of the *Verticillium* sensu stricto species is challenging and has, therefore, become the focus of extensive research [1].

Since 1990, numerous reports of molecular techniques for species identification have been reported, with the majority being based on polymerase chain reaction (PCR) [5,18,19,20,21,22,23,24]. On the basis of the differences in the internal transcribed spacer (ITS) region, primers specific to *V. albo-atrum*, *V. dahliae,* and *V. tricorpus* were designed to distinguish between the species [23,25]. Other applications of the PCR-based methodology are focused on developing pathotype-specific SCAR markers for the detection and differentiation of less virulent and highly virulent isolates [5,14,15,26]. Recently, new PCR-based simplex and multiplex assays were designed and validated for the identification and differentiation of all 10 *Verticillium* sensu stricto species and the *V. longisporum* lineages [1]. Such assays generally aim to enable accurate species identification and the diagnosis of *Verticillium* wilt diseases.

Since their discovery, more than 20 years ago, short sequence repeats (SSR) or microsatellite markers have become the most widely used markers for genotyping different species due to their ease of use. SSR markers are highly informative multi-allele codominant markers that are widely distributed throughout prokaryotic and eukaryotic genomes [26]. As a tool for detecting *V. dahliae* variability, SSR markers were developed from genomic libraries enriched with SSR repeats. The newly developed markers enabled genetic variability identification in *V. dahliae* isolates from different host plants [27,28]. Such highly polymorphic SSR markers are being used in genetic diversity studies, phylogenetic analysis, clone identification studies, and for genotypization [29].

Extensive efforts are being made to develop efficient molecular techniques for the rapid identification and differentiation of plant pathogens, which will enable the detection of specific pathogens immediately after isolation in plant fields and at low concentrations (in the early stages of infection). One such promising approach is loop-mediated isothermal amplification (LAMP), in which specific DNA is amplified at a constant temperature [30]. The usefulness of the LAMP technique as a diagnostic tool was demonstrated in several quarantine and non-quarantine plant pathogens, including viroids, bacteria, the fungal species *Botrytis cinereia,* and fungi belonging to the *Phytophtora* genus [31,32,33,34,35]. The LAMP technique was also tested as a method for the identification of different *V. dahliae* pathotypes, where primers for species differentiation were designed based on known random amplified polymorphic DNA (RAPD) markers [36]. The isolates had already been identified in the soil samples without any prior DNA purification.

The main objective of the present study was to re-classify the collection of *Verticillium* isolates according to the new taxonomic classification and nomenclature of the genus *Verticillium* sensu stricto, as suggested by Inderbitzin and coworkers [2]. First, the isolates collected in the Slovenian Institute of Hop Research and Brewing were analyzed with simplex and multiplex PCR assays, which were designed by Inderbitzin and coworkers [1], to establish their origin in accordance with the new taxonomic classification. Second, the isolates’ ITS region was sequenced, and the resulting phylogeny analyses were compared with PCR analyses to determine the compatibility with the newly introduced PCR markers. Because unambiguous identification using molecular markers is crucial for the analysis of fungi, the third objective was to test whether certain more commonly used marker systems, e.g., highly polymorphic simple sequence repeats (SSR markers), are sufficiently effective for the accurate identification of isolates. In addition, the loop-mediated isothermal amplification (LAMP) technique was also tested for its usefulness in rapid *V. nonalfalfae* species identification in the early stages of infection, which is an important factor in controlling the wilting disease of hops.

## 2. Materials and Methods

**Fungal isolates.** Overall, 104 *Verticillium* isolates from a diverse range of species and pathogenicities (highly virulent and less virulent) were included in the identification analysis. Isolates were obtained from different geographic locations in Europe, North America, and Japan and from different host plants, including hop, potato, tomato, cotton, olive, and alfalfa (Appendix B). In addition, the *Verticillium* spp. (*V. lecanii*, *V. fungicola,* and *V. nigrescens*), removed from the re-defined *Verticillium* sensu stricto genus [2], were included in the analysis. All isolates used in this study were maintained in the culture collection of the Slovenian Institute for Hop Research and Brewing, Žalec, Slovenia, as monospore cultures on potato dextrose agar at 4 °C, or stored as cultures in general fungal medium [37] in 20% glycerol at −80 °C.

**Fungal DNA extraction.** DNA was extracted from mycelia obtained from 3-day-old fungal cultures using a protoplast method, according to Bagar et al. [38]. Mycelium for DNA extraction was prepared by adding SP buffer (0.8% wt/vol NaCl, 10 mM NaH_2_PO_3_, pH 6.0) and centrifugated at 700 rpm for 5 min. The protoplasts were prepared by treating mycelia with 0.03 g/mL of lysing enzymes from *Trichoderma harzianum* (Glucanex), dissolved in KMC buffer (1 M KCl, 25 mM CaCl_2_, 10 mM MES, pH 5.8) for 3.5 h at 30 °C. Protoplasts were collected by centrifugation for 5 min at 5000 rpm at 4 °C and washed with 1 mL of STC buffer (1.2 M sorbitol, 50 mM CaCl_2_, 10 mM Tris, pH 7.5). DNA was then extracted and purified from protoplasts according to the methods described by Moller et al. [39]. DNA was re-suspended in TdE (10 mM Tris-HCl, pH 8.0 in 0.1 mM EDTA, pH 8.0). DNA concentration was determined using a fluorometer and DNA quality was assessed by 1% agarose gel electrophoresis according to the standard procedures. The isolated DNA was stored at −20 °C.

**Simplex PCR.** For species identification, according to the new taxonomic classification of the genus *Verticillium* sensu stricto, 18 new primers, combined in 11 primer pairs, were used in simplex PCR assays, developed by Inderbitzin et al. [1]. Each 20 µL reaction contained 15 µL of reaction master mix (1× PCR buffer, 2 mM MgCl_2_, 0.2 mM deoxynucleotides, 0.5 µM of each primer, 0.025 U/µL of polymerase), and 5 µL of the template DNA (diluted 1:49 with nuclease free water, regardless of the measured concentration of the extracted DNA). Amplification was conducted according to the protocols for each primer pair. Products were analyzed with gel electrophoresis with 1.3% wt/vol agarose, according to standard procedures. Each amplified PCR product of the specific isolate was evaluated by length and identified according to the expected PCR product length for each *Verticillium* sensu stricto species [2]. 

**Multiplex PCR.** For the identification and differentiation of the *V. longisporum* lineages, a multiplex PCR assay was performed according to the protocol developed by Inderbitzin et al. [1], using specific primer pairs (A1f/A1r, D1f/AlfD1r, Df/Dr, respectively). The multiplex PCR assay was used for all isolates that revealed products specific for *V. dahliae* in the simplex PCR assay and for the Species A1 and Species D1 of *V. longisporum*. Each 20 µL reaction contained 15 µL of reaction master mix (1× PCR buffer, 2 mM MgCl_2_, 0.2 mM deoxynucleotides, 0.5 µM or 0.25 µM of each primer, 0.025 U/µL polymerase) and 5 µL of template DNA (diluted 1:49). Amplification was conducted according to the multiplex PCR protocol. Products were analyzed and evaluated in the same manner as the simplex PCR products.

**ITS region amplification, sequencing, and phylogenetic analysis.** Primers ITS1 and ITS4, designed by White et al. [40], were used to amplify the fungal ITS region. Each 20 µL reaction contained 15 µL of reaction master mix (1× PCR buffer, 2 mM MgCl_2_, 0.2 mM deoxynucleotides, 0.5 µM of each primer, 0.025 U/µL polymerase) and 5 µL of template DNA (diluted 1:49). Products were analyzed with gel electrophoresis with 1.3% wt/vol agarose. 

The sequencing of the ITS region was performed by the direct sequencing of PCR products using the Sanger method [41]. The BigDye™ Terminator v3.1 Cycle Sequencing Kit (Applied Biosystems™) was used according to the manufacturer’s protocol. The samples were subsequently purified with ethanol and EDTA precipitation, following size separation by capillary electrophoresis using a 3130 XL Genetic Analyzer (Applied Biosystems™, Waltham, Massachusetts, USA). The sequencing data were analyzed using the licensed CodonCode Aligner software (ver. 5.1; CodonCode Corporation).

Phylogenetic analysis was performed using the CLC Genomic Workbench software (Qiagen). Sequences of the ITS region for 104 isolates were aligned using the MUSCLE algorithm [41] and then manually edited on the same length. A maximum likelihood phylogenetic tree was obtained using the neighbor-joining/Kimura80 model and rooted to the *Gibellulopsis nigrescens* ITS sequence (Genebank accession number NR_149327). Additionally, all sequences were compared to the NCBI database using the BLASTn tool.

**Assessing the specificity of the *V. dahliae* PCR assay.** The additional isolates of *G. nigrescens* (CBS 100826, CBS 100833, CBS 117131, CBS 120177, CBS 120949), acquired from the CBS collection, were added to the analyses to test the specificity of the Df/Dr primer pair for the identification of *V. dahliae* isolates [1], as the first PCR experiment resulted in some unspecific amplifications in the *G. nigrescens* strains using the *V. dahliae*-specific primers (Df/Dr primer pair). PCR amplification was performed according to the protocol designed by Inderbitzin et al. [1]. The amplification products were analyzed and evaluated in the same manner as the simplex PCR products.

**SSR marker analysis.** Sixty-one new specific microsatellite markers were developed de novo in order to analyze the genetic diversity of the collection of *Verticillium* spp. isolates. Sequences of *V. alfalfae* (VaMs.102; GenBank assembly accession: GCA_000150825.1) were used to search for the presence of microsatellites using MISA (MIcroSAtellite identification tool) [42] with the default parameters for identifying SSR repeats. The Primer3 software, which is freely available online, was used to design the specific primers from the flanking regions. The search was executed on the whole genome, genes, transcripts, and mitochondrial sequences. Sixty-one pairs of newly designed primers, elongated for the universal M13 tail sequence [43], were tested on 16 selected fungal isolates (P10, CD15, T2, Rec91, 11100, 298092, 102.464, 11066, 11077, 11081, 107, ARO1JS1, Pd83-53A, JKG2, MAI, PAP2008) of *V. alfalafae*, *V. nonalfalfae*, *V. albo-atrum,* and *V. dahliae*. Thereafter, 12 primer pairs with the highest obtained polymorphism were chosen for further analyses of all collected isolates.

**Microsatellite amplification and loci analysis.** The PCR amplification of 12 SSR loci was performed in 15 µL reactions (1× PCR buffer, 2 mM MgCl_2_, 0.2 µM of each deoxynucleotide, 2.5 mM specific forward and reverse primer, 0.25 µM TAIL M13 primer (5′d[GTAAAACGACGGCCAG]3′), 1.25 U/reaction GoTaq DNA polymerase) with 20 ng of fungal DNA. The reactions were incubated in a thermal cycler as follows: initial denaturation at 94 °C for 5 min, followed by five cycles at 94 °C for 45 s, and annealing at 60 °C for 30 s with elongation at 72 °C for 1 min. For each cycle, the temperature was reduced by 1 °C. Annealing was followed by 25 cycles at 55 °C, and the final step was elongation at 72 °C for 8 min. 

The PCR fragments with fluorescent FAM-, VIC-, NED-, and PET-labelled microsatellite primers were combined and separated using a 16-capillary 3130 xl genetic analyzer (Applied Biosystems). The sizes of the PCR products were analyzed using GeneMapper 4.0 (Applied Biosystems). The allele sizes of specific isolates were clustered using the microsatellite analyzer (MSA) 4.05 software package [43] by calculating the Dps (proportion of shared alleles) distance coefficients. These were used as a base for the dendrogram construction using the neighbor-joining algorithm, employing the PHYLIP software package (3.6b version; University of Washington). The visualization of the clusters was achieved with the MEGA7 software (version 7.0) [44].

**LAMP PCR primer design and analysis.** The *V. nonalfalfae*-specific primers for the LAMP assay were designed using the Primer Explorer V5 software. Firstly, the genome sequences of the *V. dahliae* isolate Vdls17 (GenBank assembly accession: GCA_000150675.2) and the *V. nonalfalfae* isolate T2 (GenBank assembly accession: GCA_002776445.1) were aligned, and the sequence regions only present in *V. nonalfalfae* were selected for primer design. The genomic regions only present in *V. nonalfalfae* were divided into three parts and different sets of primer pairs were developed to differentiate between *V. nonalfalfae* and *V. dahliae*. To differentiate between the less virulent and highly virulent *V. nonalfalfae* isolates, two sets of LAMP primers (namely, PG2_1 and PG2_2) were designed in the lethal region of the highly virulent *V. nonalfalfae* isolate T2, which was previously discovered by Jakše et al. [45]. Thus, we only amplified the DNA of the highly virulent isolates. The primer pairs selected for the LAMP analysis originated from the regions where the maximum number of sequence differences between the two species/pathotypes was observed in the genome alignments. In the first step, the F3/B3 and FIP/BIP primer pairs were retrieved, and in the second step, the LoopF and LoopB primers were designed based on the previously selected F3/B3 and FIP/BIP primer pairs using the Primer Explorer V5 [30].

The LAMP reaction was performed using the WarmStart LAMP Kit (NEB). The designed primers were diluted to 10 nM concentrations and 2.5 µL of a specific primer pair was used in a LAMP reaction, supplemented with 12.5 µL of the 2× WarmStart Master Mix and 2 µL of fungal DNA for a total amount of 2 ng of DNA. The temperature profile used for the amplification was 45 min at a constant temperature of 65 °C. If needed, an optimized temperature profile (68 °C for 5 min, 67 °C for 5 min, 66 °C for 5 min, and 65 °C for 30 min) was used for the increased specificity of the amplification. We selected 16 *Verticillium* sensu stricto isolates (Appendix B) as the representatives of each species to be included in the LAMP analysis. The visualization of the LAMP reactions was conducted using 2% E-Gel Precast Agarose Gels (ThermoFisher, Waltham, Massachusetts, USA). For each reaction, 12 µL of sample, supplemented with brome phenol blue dye (NEB), was used and loaded onto the gels. The electrophoresis was held in an iBase device (ThermoFisher) for 45 min. The reactions were visualized using the UV transilluminator and recorded using the VisionWorks software.

## 3. Results

### 3.1. Simplex and Multiplex PCR Assays 

In total, 88 isolates out of 104 were identified with the newly designed species-specific simplex PCR assays [1], according to PCR products of specific length (Appendix B). Thirteen isolates were not identified due to poor DNA amplification. The *V. klebahnii* and *V. zaregamsianum* species were not identified among the analyzed samples with the introduced PCR marker approach or by the sequencing of the ITS region due to the absence of the two species in the fungal collection. 

Thereafter, multiplex PCR for the 35 selected isolates that exhibited positive amplification with the Df/Dr primer pair and in the *V. longisporum* assays was conducted using the *V. dahliae*–*V. longisporum* assay to distinguish between *V. dahliae* and different *V. longisporum* lineages. We confirmed the identity of the *V. longisporum* A1/D1 lineage for two isolates (PD330, CBS 110.218) (Appendix B), specified by the *V. longisporum* species A1 and *V. longisporum* species D1 PCR amplicons. The remaining isolates of *V. dahliae* (namely CIG3, JKG 2, JKG1, JKG 8, DJK, MAI, Mint, GAJ09, PDRENU/MAR, CasD, KresD, MoD, Oset, 12042, PAPmb, PAP, Pap99, Pap2008, 14, 141, 3V, 802-1, V 138 I, V 176 I, PD335, PD584, A III 25, PD337) were confirmed using the previously described simplex PCR identification, as the *V. dahliae*-specific amplicon was detected. 

### 3.2. Phylogenetic Analysis Based on ITS Sequencing

On the basis of the obtained sequence data for the ITS region, a total of 100 isolates were included in the phylogenetic analysis. *Verticillium nonalfalfae* isolate 11047 and *V. dahliae* 802-1 were excluded from the analysis because low quality ITS sequences were obtained. The same applied to isolates representing *V. fungicola* and *V. leicanii* species, the two species removed from the *Verticillium* sensu stricto genus (Appendix B). A maximum likelihood phylogeny was obtained using the neighbor-joining/Kimura80 model, which divided all 98 isolates into 12 groups/branches (Table 1), represented by 98 isolates of the *Verticillium* sensu stricto species, together with their specific sub-groups, and two isolates of the *G. nigrescens* species. The consensus sequence of each group was used in the alignment of sequences and the final phylogenetic tree design (Figure 1). The bootstrap value of 100% showed that the isolates of *Verticillium* sensu stricto were grouped together in a clade in all performed bootstrap replications with a clear distinction from *G. nigrescens* isolates.

The branches of the phylogenetic tree well represent specific *Verticillium* sensu stricto species, and the identification based on ITS sequences is in accordance with the simplex/multiplex PCR assay identification (Appendix B). Interestingly, for the *V. albo-atrum* species, sub-clustering was observed, as isolate CBS 102.464 (UK, artichoke isolate) appeared to be different from the other five isolates. The same sub-division was observed with the SSR marker analysis, where the CBS 102.464 isolate had two polymorphic loci (235 bp long locus no. 2756 and 193 bp long locus no. 3507) with longer alleles than the other *V. albo-atrum* isolates (222 bp and 187 bp). The results indicate that the specific isolate from artichoke underwent some type of evolutionary adaptation that might be host-dependent. 

### 3.3. Specificity of Verticillium dahliae Simplex PCR Assay

A total of 42 isolates were included in the specificity test, including 27 *V. dahliae* isolates, 7 *G. nigrescens* isolates, 2 *V. longisporum* A1/D1 isolates, and 1 of each of the *V. albo-atrum*, *V. alfalfae*, *V. nubilum*, *V. tricorpus*, *V. isaacii*, and *V. nonalfalfae* isolates. The Verticillium dahliae-specific marker (defined by the Df/Dr primer pair [1]) was amplified in all *V. dahliae* isolates, six *G. nigrescens* isolates, one *V. longisporum* isolate, and one *V. isaacii* isolate. The length of the amplicon (490 bp) was specific to the Df/Dr amplicon, typical for the *V. dahliae* species (Figure 2).

### 3.4. SSR Marker Development

A total of 628 sequences of the V. alfalfae isolate VaMs.102 contained SSRs, wherein the most frequent were dinucleotides (52.9%) and trinucleotides (30.4%). The search was conducted on the whole genome, genes, transcripts, and mitochondria. Nine microsatellite repeats were found in mitochondria, 170 were found in genes, and, out of these, 104 were also confirmed in transcripts. The rest of the microsatellites were in non-coding regions. Out of the 104 microsatellite sequences, 61 were chosen for primer development based on the length of the repeating region, the low GC content of the repeat, and their uniform distribution along the whole sequence of *V. alfalfae*. PCR amplification was not successful for two of them and eight only showed amplification for *V. alfalfae* isolates and were therefore omitted from the analysis. The remaining 54 primer pairs successfully amplified specific products in each of the 16 isolates preliminarily tested, resulting in between two and nine amplified alleles per locus. Twelve primer pairs, with the highest polymorphic information content, were chosen for further genetic/similarity analysis (Table 2). 

Ninety-four genotypes/*Verticillium* sensu stricto isolates, analyzed for twelve SSR markers, exhibited a total amplification of 54 alleles. Between two and nine alleles were amplified per locus (Appendix A); the highest polymorphism was observed for locus 1566 with nine alleles and the lowest for locus 1468 with only two alleles. For the locus designated as 3632, the 237 bp long allele was detected in all included *V. nonalfalfae* species, while the same locus produced a 252 bp long allele in the *V. alfalfae* species and a 235 bp long product in the *V. dahliae* species. Another locus, marked 886, produced three alleles, wherein the 168 bp long product was only observed in the *V. dahliae* species. In *V. alfalfae*, a specific 213 bp long allele was detected for locus 3111, differentiating the *V. alfalfae* and *V. nonalfalfae* species, and, as for the latter, a 194 bp long allele was observed. In addition, in *V. nubilum*, *V. fungicola*, and *V. longisporum*, alleles of a specific length were detected for loci 3632, 886 2390, and 1556, respectively. In the case of *V. albo-atrum*, several loci (598, 2756, and 3507) produced alleles of a specific length and, thus, need to be used in combination to differentiate the species. The same applies to *V. isaacii*, wherein the loci designated as 959, 228, and 1556 enable differentiation of the species among other *Verticillium* sensu stricto isolates.

The phylogenetic tree constructed using the proportion of shared alleles coefficient resulted in a dendrogram containing four distinct clades (Appendix A), representing the different *Verticillium* sensu stricto species included in the analysis. The biggest group represented *V. nonalfalfae* isolates, which is in accordance with the ITS marker analysis and phylogeny. All included isolates were identified as *V. nonalfalfae* in all three identification approaches. The second largest clade represents *V. dahliae* species, which was expected, as the collection of the analyzed isolates contained a significant number of *V. dahliae* isolates. The *V. alfalfa* clade and *V. longisporum* clade contain isolates that were also identified as such in the ITS marker phylogeny. 

### 3.5. LAMP Sensitivity for Species Identification

Four sets of LAMP primers for the identification and differentiation of *V. nonalfalfae* species from the *V. dahliae*, *V. alfalfae,* and *V. albo-atrum* species were developed (namely Vna_Vd1_1, Vna_Vd1_2, Vna_Vd2, and Vd-mitoh), together with two LAMP primer pairs for *V. nonalfalfae* pathotype identification (Appendix A). The usefulness of the designed primers was tested on selected *Verticillium* sensu stricto isolates. 

The Vna-Vd1_1 primer pair resulted in the specificity of the amplification after protocol optimization, wherein the LoopF and LoopB primers were excluded from the reaction, as their use was not mandatory in the LAMP reaction. After optimization, we only observed positive reactions in *V. nonalfalfae* isolates, whereas for *V. dahliae* isolates, DNA was not amplified (Figure 3), indicating that the designed primer pair Vna-Vd_1 differentiated between *V. nonalfalfae* and *V. dahliae*. The Vna_Vd1_2 primer pair differentiated between the *V. nonalfalfae* and *V. dahliae* isolates, and no amplification was observed for the *V. dahliae* isolates; however, the DNA failed to be amplified in several *V. nonalfalfae* isolates (Figure 4).

Thereafter, the Vna_Vd2 primers resulted in the amplification of the *V. nonalfalfae* and *V. dahliae* isolates, and no DNA amplification was observed for the *V. alfalfae* and *V. albo-atrum* isolates (Figure 5). The primer pair thus enables the exclusion of the *V. albo-atrum* and *V. alfalfae* species when analyzing unknown isolates. By using the Vd-mitoh primer pair, designed in the *V. dahliae* mitochondrial sequence, only DNA from the *V. dahliae* isolates was amplified (Figure 6); however, the DNA was not amplified in all investigated *V. dahlia*e samples, as expected.

The PG2_1 LAMP primers only resulted in positive amplifications in the highly virulent *V. nonalfalfae* isolates (Figure 7). No DNA was amplified in the less virulent *V. nonalfalfae* isolates. However, DNA was amplified in the *V. dahliae* isolates. These results were not unexpected, as the lethal DNA regions of *V. dahliae* and *V. nonalfalfae* share a common origin [5]. On the other hand, PG2_2 primers also resulted in the amplification of the less virulent *V. nonalfalfae* isolates, meaning that no pathotype-specificity was observed. 

For accurate species identification, the combination of newly designed 1) Vna_Vd2 primers for the differentiation of *V. alfalfae* and *V. albo-atrum* species from *V. nonalfalfae* and *V. dahliae*, 2) an optimized Vna_Vd1_1 and Vna_Vd1_2 protocol for the differentiation of *V. nonalfalfae* from *V. dahliae,* and 3) PG2_1 primers for the differentiation of the highly virulent and less virulent pathotypes of *V. nonalfalfae* were necessary to determine the *V. nonalfalfae* isolates. The listed primer combinations, used in specific order, enable the fast diagnostic analysis of unknown *Verticillium* sensu stricto isolates, which are primarily hosted by hop (*Humulus lupulus* L.)

## 4. Discussion

The genus *Verticillium* sensu stricto, established in 2011, comprises 10 plant-pathogenic species differing in pathogenicity and host range [2]. As a result of the possible changes of the morphology in laboratory conditions, the identification of individual species based on morphology and pathogenicity tests remains difficult [1,17]. Inderbitzin et al. (2013) designed simplex and multiplex PCR assays as a tool to identify all 10 *Verticillium* sensu stricto species on a molecular level. Molecular techniques enable the quick and accurate identification of fungi and allow for the diagnosis verticillium wilt disease [1,3,5,6].

In our study, we identified and analyzed the diversity of *Verticillium* spp. isolates from the Slovenian Institute of Hop Research and Brewing using several different molecular-based approaches. Analyses using new simplex and multiplex PCR assays [1] enabled us to accurately identify 88 isolates. Fungal isolates retrieved from infected hop plants were identified as *V. nonalfalfae*, a species that commonly infects hop, whereas isolates obtained from infected lucerne seedlings were identified as *V. alfalfae* (Appendix B), a species that only infects lucerne [2]. Both species derive from the previously established Grp I group of *V. albo-atrum*, which is sub-divided into lucerne and non-lucerne pathogenic fungi [3,12,13,23,46,47,48]. Since the new taxonomic classification of the genus *Verticillium* sensu stricto in 2011, the name *V. albo-atrum* now refers only to the previously established Grp II group of *V. albo-atrum* [3,23]. With the simplex PCR assay, using a *V. albo-atrum*-specific primer pair, we successfully identified and re-classified six representatives of the re-defined *V. albo-atrum* species (Appendix B). 

In 2011, Inderbitzin et al. divided *V. tricorpus* into three different species (*V. klebahnii*, *V. isaacii,* and *V. tricorpus*) based on multiple loci phylogenetic analysis. According to the new classification, we identified two *V. tricorpus* isolates and three *V. isaacii* isolates (Appendix B) that were previously all assigned as *V. tricorpus*. In contrast, isolate 115 was morphologically identified as *V. albo-atrum*, but the molecular analysis revealed that this isolate is in fact of a *V. isaacii* species. It is our belief that the incorrect morphological identification was due to the isolate being isolated from potato, which is the common host of *V. albo-atrum*. Thus, the species have a similar morphology, e.g., forming resting mycelia, microsclerotium, and yellow pigmented hyphae [2,48].

Among all tested isolates, 28 were identified as *V. dahliae* based on the *V. dahliae*-specific simplex PCR assay using the Df/Dr primer pair (Appendix B); however, three of them were later identified as *G. nigrescens* in the ITS region-based phylogeny. Thus, we performed additional testing on some other *G. nigrescens* isolates obtained from the CBS collection. In the additional specificity testing, we also included other *Verticillium* sensu stricto species. PCR products of specific length were confirmed in the control group, comprising the three *V. dahliae* isolates, in all but one of the *G. nigrescens* isolates, in one out of two of the *V. longisporum* A1/D1 isolates, and in one *V. isaacii* isolate (Figure 2). As expected, amplification was not observed in the *V. albo-atrum*, *V. alfalfa*e, *V. nubilum*, *V. tricorpus,* and *V. nonalfalfae* isolates. According to these results, the simplex PCR assay for *V. dahliae* identification based on the ITS target locus with the originally proposed protocol [1] needs further examination and optimization. The *G. nigrescens* isolates, which were predominantly positive using this assay, have similar hosts (such as cotton, potato, rape) as the *V. dahliae* species [2,7,48], increasing the risk of false positive diagnoses. 

ITS region amplification and direct sequencing using ITS1 and ITS4 primers [49] resulted in 11 distinct groups, representing different *Verticillium* sensu stricto species and *G. nigrescens* isolates. The phylogenetic tree (Figure 1) represents the relationships between *Verticillium* sensu stricto species and *G. nigrescens* isolates. With this approach, we were able to accurately identify all isolates included in the analysis. Inderbitzin et al. (2011) obtained comparable results from their phylogenetic tree based on the ITS dataset, reporting the same relationships between the species as in our study. In both trees, the characteristic distribution into two clusters was observed, one being Flavexudans and the other Flavnonexudans. In addition, we identified sub-groups within the species *V. albo-atrum* and *V. dahliae* (Figure 1). The *Verticillium albo-atrum* sub-group (CBS 102.464), which differed from other *V. albo-atrum* isolates, has artichoke as its host plant. In the *V. dahliae* sub-group, the isolates differed in pathogenicity and origin. Similarly, Collopy et al. (2002) identified sub-groups in *V. fungicola* isolates, wherein the sub-grouping correlated with the geographical location or origin of the isolates [46,48]. In our study, we considered the geographical origin, host plant, and pathogenicity of the isolates. 

With the SSR marker analysis, we confirmed the expected bias related to the approach of developing SSR sequences: the repeating regions of microsatellites are usually longer in the species from which they are developed and shorter in other related species, a phenomenon known as ascertainment bias [50]. Our results showed that alleles from the *V. alfalfae* isolate were longer (157 bp for locus 1566 and 243 bp for locus 1413) than those from *V. nonalfalfae* (111 bp for locus 1566 and 198 bp for locus 1413), which is in accordance with the hypothesis proposing that microsatellite loci identified by DNA library screens over-represent the repeat size distribution in a given species’ genome because the screening conditions favor the identification of clones with long repeat units [51,52]. Since homologous loci have different evolutionary histories within different species, long microsatellite loci in one species will not necessarily be long in another species. Thus, if primers developed from one species are used to amplify microsatellites in other species, length ascertainment bias can result in an observed difference between species [50]. Moreover, the length constraint of the microsatellite region is known to have an important impact on SSR variability, and the species with longer microsatellites should exhibit higher levels of population variation, which was confirmed in our study. 

The results of the comparative analysis of variability among the three groups of different *Verticillium* sensu stricto species show that the calculated average diversity was almost seven times higher in the *V. alfalfae* population than the *V. nonalfalfae* population and three times higher than the *V. dahliae* population (Appendix A). However, microsatellite markers accurately identified 86 *Verticillium* sensu stricto isolates (Appendix A). The species-specific identification was in accordance with the ITS identification, making SSR marker-based phylogeny a suitable method for accurate species determination. Moreover, as the developed SSR markers differ in length among species, they could be used for the development of species-specific assays for rapid isolate identification. The development of a *V. dahliae*-specific assay using the SSR loci 886 could potentially overcome the lack of specificity of the Df/Dr primer pair of the *V. dahliae* simplex PCR assay [1]. 

The LAMP approach [30], which allows for rapid identification, early diagnostics, and identification without prior DNA purification, was demonstrated to be useful for the identification of the *Verticillium* sensu stricto species. The LAMP primers were developed using the *V. nonalfalfae* genome [45] and *V. dahliae* genome [48], as these are the two most problematic species in hop production. The LAMP primers were developed in regions of the *V. nonalfalfae* genome that share no homology with the *V. dahliae* genome and in the lethal-specific region of the highly virulent *V. nonalfalfae* strain that shares no homology with the less virulent *V. nonalfalfae* strain [53]. However, the amplification of the LAMP PCR products, using the newly developed Vna_Vd1_1 and Vna_Vd2 primer pairs, was also observed in the *V. dahliae* isolates PAP2008 and CasD. After protocol optimization, the Vna_Vd1_1 primer pair only amplified the *V. nonalfalfae* isolates, and, as expected, no PCR products were detected in the *V. dahliae* species (Figure 3). When analyzed in combination with the Vna_Vd1_2 primer pair, the LAMP assay enabled the accurate identification of all *V. nonalfalfae* species and differentiation from the *V. dahliae* species. We discovered that the variability of the selected regions is high, probably due to either horizontal gene transfer or poor sequence assemblies in the genomic project (Appendix A), resulting in some unexpected amplifications for certain isolates. We also believe that the variability might be the consequence of species adaptation to hosts, the environment, and the geographical location [3,48]. However, the developed LAMP PCR assays, when used simultaneously (Vna_Vd2, the optimized Vna_Vd1_1 protocol, and the Vna_Vd1_2 primer pair together with the Vd_mitoh primer pair), enable accurate *V. nonalfalfae* identification. 

## 5. Conclusions

*Verticillium* sensu stricto is a diverse group of plant-pathogenic fungi comprising 10 species that are differentiated by morphological characteristics, host plants, and geographical origin. In the present study, a collection of *Verticillium* isolates was re-classified with newly developed simplex and multiplex PCR assays, which enabled the accurate molecular identification of all 10 species. With this approach, 88 isolates from the collection were successfully identified, and the specific species were additionally confirmed using the ITS region-based phylogeny. However, during the identification analysis, the specificity of the *V. dahliae* simplex PCR assay was questionable, as the PCR products were also detected in other species (*G. nigrescens*, *V. longisporum*, *V. isaacii*). Therefore, the Df/Dr primer pair should be evaluated and re-designed to only amplify the *V. dahliae*-specific ITS region. The collection of *Verticillium* sensu stricto isolates was also used in the SSR marker analysis and phylogeny, which were conducted using 12 newly designed highly polymorphic SSR markers found in the *V. alfalfae* genome. SSR loci of specific length or their combinations were determined in each *Verticillium* sensu stricto species, thus enabling the development of species-specific markers for isolate identification. Lastly, the LAMP technique was tested for use as a rapid and cost-effective species identification method, with a focus on the *V. nonalfalfae* and *V. dahliae* species. Despite the fact that the primers were developed in *V. nonalfalfae*-specific or *V. dahliae*-specific genomic regions, amplification was not as specific as expected. We believe that this is due to species’ adaptation to specific environments/niches, which can be observed on the genomic level. Therefore, further studies are needed to develop accurate LAMP-based identification protocols for *Verticillium* sensu stricto species. 

## Figures and Tables

**Figure 1 pathogens-12-00535-f001:**
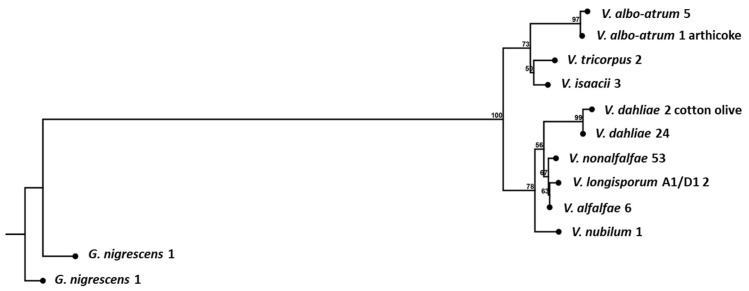
Maximum likelihood phylogenetic tree constructed on ITS sequences for the 98 *Verticillium* sensu stricto isolates; branches represent the number of isolates with the same ITS sequence; the names of the isolates grouped in a specific branch are provided in Table 1.

**Figure 2 pathogens-12-00535-f002:**
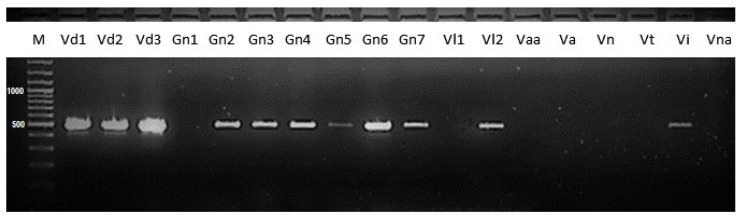
Results of the specificity test for *V. dahliae* specific PCR assay, including representatives of each species. M—1 kilo bp marker; Vd1—*V. dahliae* AIII25; Vd2—*V. dahliae* PD337; Vd3—*V. dahliae* CIG3; Gn1—*G. nigrescens* PAPmb; Gn2—*G. nigrescens* CBS 123.176; Gn3—*G. nigrescens* CBS 100826; Gn4—*G. nigrescens* CBS100833; Gn5—*G. nigrescens* CBS 120949; Gn6—*G. nigrescens* CBS 120177; Gn7—*G. nigrescens* CBS 117131; Vl1—*V. longisporum* A1/D1 CBS 110218; Vl2—*V. longisporum* A1/D1 PD330; Vaa—*V. albo-atrum* 112; Va—*V. alfalfae* Luc; Vn—*V. nubilum* CBS 456.51; Vt—*V. tricorpus* CBS 227.84; Vi—*V. isaacii* JKG20; Vna—*V. nonalfalfae* TABOR6; M—100 bp gene ruler.

**Figure 3 pathogens-12-00535-f003:**
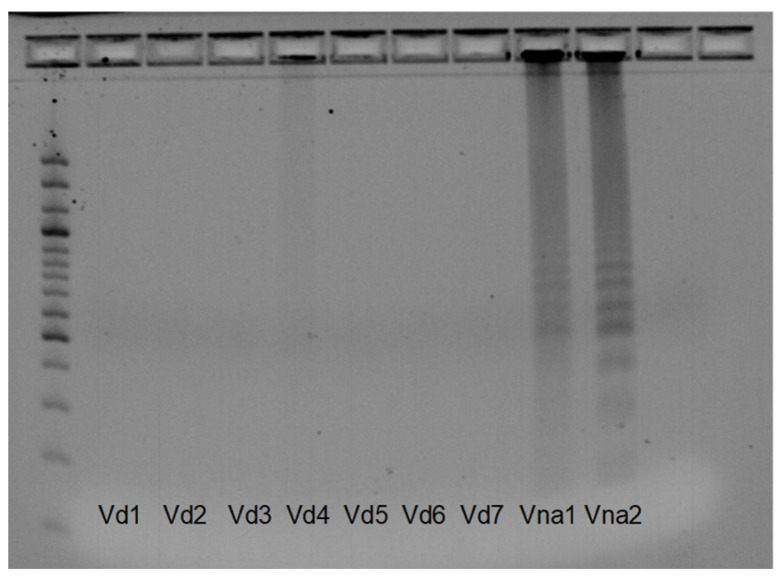
Results of the optimized LAMP reaction using the Vna_Vd1_1 primer pair. *V. dahliae* isolates: Vd1—Mint; Vd2—Oset; Vd3—PAP 1999; Vd4—PAP2008; Vd5—JKG2; Vd6—KresD; Vd7 **—**CasD; Vna1—less virulent *V. nonalfalfae* isolate; Vna2—highly virulent *V. nonalfalfae* isolate; size standard—100 bp.

**Figure 4 pathogens-12-00535-f004:**
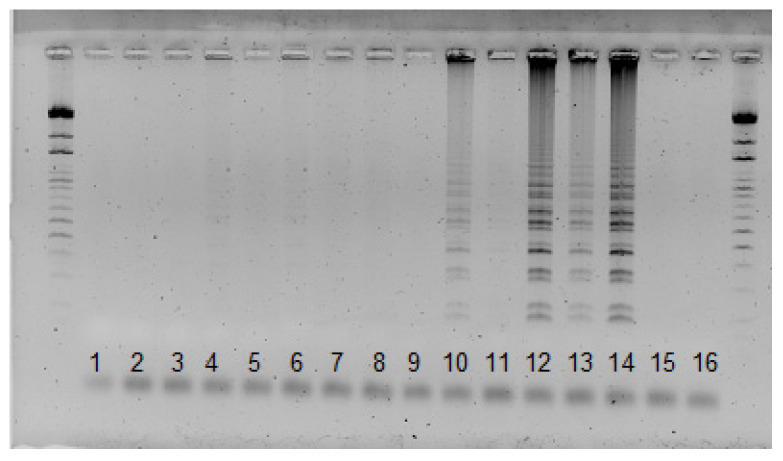
Results of the LAMP reaction using the Vna_Vd1_2 primer pair. As shown, 1, 2—*V. albo-atrum* isolates 110 and 112; 3, 4—*V. alfalfae* isolates 107 and 11; 5–7—less virulent *V. nonalfalfae* isolates Zup, Rec91, and KRES 98; 8–13—highly virulent *V. nonalfalfae* isolates 1985, 11100, T2, BIZ, P 15, and P 10; 14—less virulent *V. nonalfalfae* isolate P83; 15, 16—*V. dahliae* isolates PAP2008 and CasD; size standard—100 bp.

**Figure 5 pathogens-12-00535-f005:**
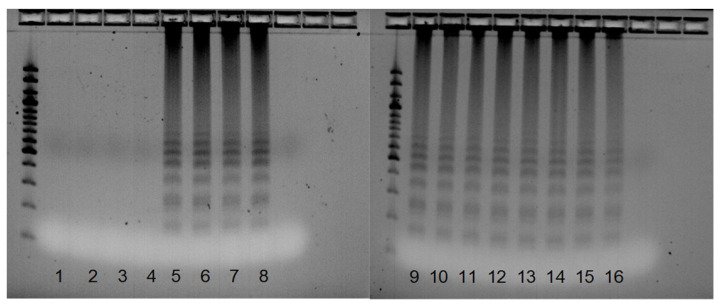
Results of the LAMP reaction using the Vna_Vd2 primer pair. As shown, 1, 2—*V. albo-atrum* isolates 110 and 112; 3, 4—*V. alfalfae* isolates 107 and 11; 5–7—less virulent *V. nonalfalfae* isolates Zup, Rec91, and KRES 98; 8–13—highly virulent *V. nonalfalfae* isolates 1985, 11100, T2, BIZ, P 15, and P 10; 14—less virulent *V. nonalfalfae* isolate P83; 15, 16—*V. dahliae* isolates PAP2008 and CasD; size standard—100 bp.

**Figure 6 pathogens-12-00535-f006:**
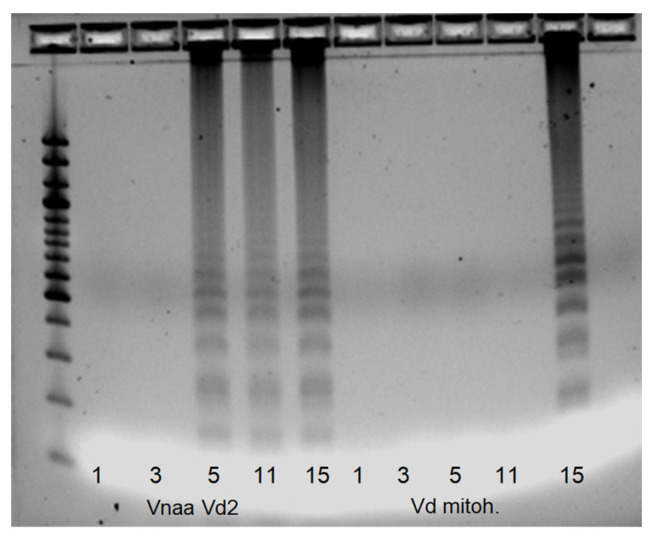
Results of the LAMP reaction using the Vd-mitoh primer pair. As shown, 1—*V. albo-atrum* isolate 110; 3—*V. alfalfae* isolate 107; 5—less virulent *V. nonalfalfae* isolate Zup; 11—highly virulent *V. nonalfalfae* isolate BIZ; 15—*V. dahliae* isolate PAP2008; size standard—1 kilo bp.

**Figure 7 pathogens-12-00535-f007:**
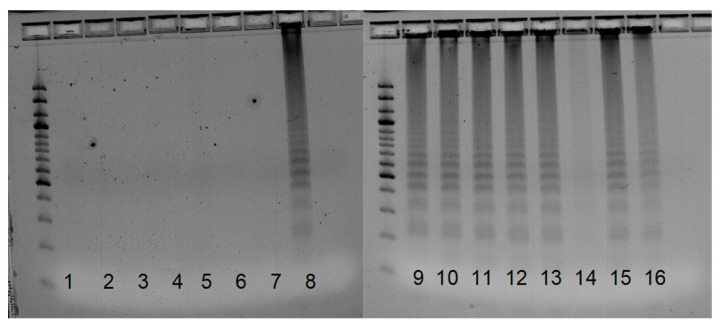
Results of the LAMP reaction using the PG2_1 primer pair. 1, 2—*V. albo-atrum* isolates 110 and 112; 3, 4—*V. alfalfae* isolates 107 and 11; 5–7—less virulent *V. nonalfalfae* isolates Zup, Rec91, and KRES 98; 8–13—highly virulent *V. nonalfalfae* isolates 1985, 11100, T2, BIZ, P 15, and P 10; 14—less virulent *V. nonalfalfae* isolate P83; 15, 16—*V. dahliae* isolates PAP2008 and CasD; size standard—100 bp.

**Table 1 pathogens-12-00535-t001:** Verticillium sensu stricto species, forming the branches/groups in a phylogenetic tree.

Branch Designation	Isolates Forming Groups with the Same ITS Sequence
*V. alboatrum* 5	110, 112, 166, PD693, CBS 682.88
*V. alboatrum* artichoke 1	CBS 102.464
*V. isaacii* 3	115, EX5 F8, JKG 20
*V. tricorpus* 2	EX5 F7, CBS 227.84
*V. dahliae* 2 cotton olive	14, V-176 I
*V. dahliae* 24	AIII25, PD337, JKG2, Mint, KresD, CasD, CIG3, 3V, V-138 I, Pap99, Pap2008, PDRENU, PD584, PD335, PAP, Oset, MoD, MAI, JKG8, JKG1, GAJ09, DJK, 141, 12042
*V. nonalfalfae* 53	340646, 314139, 298101, 298102, 298100, 298092, 1985a, 15/99, 11097, 11081, 11077, 11066, 11055, 11041, P10, P114/1, KRES98, 1953, 14/93, KV11, Zup, VranBis09, T-179, TABOR6, T2, Surf, Sol, Sent04, Rec91, PETROL, PD83/53a, P84/2, P34/1, P15, OCer, MO 3, PD2000/4186a, Kum, Ciz, CBS 454.51, CBS 393.91, CBS 321.91, BIZ, AR0/140, AR01/JS1, AR01/067
*V. longisporum* A1_D1 2	PD330, CBS 110.218
*V. alfalfae* 5	Luc, Kanada11, 107, 41, CBS 392.91
*V. nubilum* 1	CBS 456.51
*G. nigrescens* 1	PapMB
*G. nigrescens* 1	CBS 123.176

**Table 2 pathogens-12-00535-t002:** Twelve newly designed SSR markers with the highest polymorphism used for Verticillium sensu stricto species differentiation.

SSR Locus Designation	Motif	Length (bp)	Forward Primer 5′-3′	Reverse Primer 5′-3′
598	(AG)16	177	TGTGAGGGCACTGACATGAT	AGAACAGCCTCTTCCGTCAA
959	(GA)19	213	CCAACCCTTCCATCACATTT	TGGAGGCAGTGATGAGTTTG
228	(AGA)14	200	CCGGACGAGAGAGTCTGAAG	TTTGAGCAGATTATGCGTCG
104	(CT)21	188	CCCTTTTCCCATCTTCAACA	TGACCGAGGAGGAGTTTGAC
886	(TG)25	160	CAGCAAGGAAGCACTGTCAA	TCCAGACTACACTCTCGCCC
2756	(GT)17	226	TACGATGCGCTCTCAGAATG	CTCCTGTCAGAAGGTCCAGC
3507	(GA)17	197	AGAGGATGGCATGTCTGGAT	AGACAGTCTGCCTTGCCAAT
1468	(TCT)10	171	GATGCGGTTCTTTGTGGACT	CAAAGGGTCATGGTGTCAGA
2390	(CT)17	145	CCATGGTCGTATCTCGTGTG	GGCGGCTCTACTTCAATCAG
3111	(GA)24	200	TGTAAGCTTTGCGTGACCTG	CCCTGAGCCAATTTATCGAA
1566	(CT)16	130	TCGGATCCCAGGAGTAAGTTT	GCACGAGCTGGAAATTCTGT
3632	(GA)15	230	GACGTGGAGAAGGTGGAGAG	GCCGCCTATGTAAGCAAAGA

## Data Availability

Not applicable.

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
