# Peer review of "Novel Identification of the Collection of Pathogenic Fungal Species Verticillium with the Development of Species-Specific SSR Markers"

_pathogens, 2023, doi:10.3390/pathogens12040535_

Round 1

Reviewer 1 Report

Dear authore,

Thank for the nice article.

This manuscript is described about reclassification and identification  fungi in Verticillium based on based on PCR Markers, SSR Marker analysis, and LAMP technique.

Minor correction and validation are required. Please check the attachment.

Best wishes,

Author Response

Dear reviewer,

Thank you wery much for your positive opinion and suggestions over our manuscript.

Best wishes, Nataša Štajner

Reviewer 2 Report

Verticillium dahliae (Vd) is one of the most important soil-borne fungi which could cause about 400 crop species wilt. In this study, specific PCR markers, SSR marker analysis and LAMP technique was applied to identify the Verticillium species. Some details should be revised. Below some minor suggestions are listed:

1. Please include specific results in the summary section.

2. The background in Figure 2 is cluttered. It is suggested to replace the picture.

3. line 245, There are no concrete results.

4. Please add the length of marker to the Figure 2, 3 and 4.

5. In Figure 1, two branches are too far apart from the other branches. It is suggested to reconstruct the evolutionary tree after processing the sequence.

6. Please note the case of Latin names. For example, "verticillium" should be capitalized and italicized in line 66.

7. Please use punctuation accurately. For example, in line115, "0,8" should be changed to “0.8”.

8. The background of the protein electrophoresis figure is cluttered. Please process the image.

9. Please check the reference format carefully. “verticillium” is no italicization in line 575.

10. Please unify the units of the full text. For example, in line 86 “15 µl”, but line 219 is “2,5 µL”.

Author Response

(The authors gave the same response as above.)

Round 2

Reviewer 2 Report

Now, the quality has been improved. However, two details must be revised. 

1. The size of marker should be added in the figure 2-4, rather than legend. 

2. line 146, "0.25µM" should be "0.25 µM".